# Fuzzy-Logic-Based, Obstacle Information-Aided Multiple-Model Target Tracking

**Quanhui Wang [1,\*], En Fan [2] and Pengfei Li [3]**

1   School of Information Engineering, Lingnan Normal University, Zhanjiang 524000, China
2   College of Mechanical and Electrical Engineering, Shaoxing University, Shaoxing 312000, China; efan@szu.edu.cn
3   Chinese PLA Army Artillery Air Defense Academy Zhengzhou Campus, Zhengzhou 450000, China; roc5683@szu.edu.cn
*   Correspondence: wangquanhui@lingnan.edu.cn

**Abstract:** Incorporating obstacle information into maneuvering target-tracking algorithms may lead to a better performance when the target when the target maneuver is caused by avoiding collision with obstacles. In this paper, we propose a fuzzy-logic-based method incorporating new obstacle information into the interacting multiple-model (IMM) algorithm (FOIA-MM). We use convex polygons to describe the obstacles and then extract the distance from and the field angle of these obstacle convex polygons to the predicted target position as obstacle information. This information is fed to two fuzzy logic inference systems; one system outputs the model weights to their probabilities, the other yields the expected sojourn time of the models for the transition probability matrix assignment. Finally, simulation experiments and an Unmanned Aerial Vehicle experiment are carried out to demonstrate the efficiency and effectiveness of the proposed algorithm.

**Keywords:** target tracking; multiple model estimation; obstacle information; fuzzy inference

## 1. Introduction

The Multiple-Model (MM) algorithm is an effective approach to maneuvering target tracking in many real-world applications [1], that works by regarding the maneuvering as the transition to motion modes and describes them by a finite number of kinematic models. The MM algorithm can be categorized into three generations: autonomous, cooperative, and variable-structured. For the first generation, all model filters in a fixed-structure model set work independently without interacting with each other. Its representative is the Static Multiple Model (SMM) algorithm pioneered by Magill [2]. The second generation reinitializes each filter with a weighted sum of the updated estimates from every model in the set and merges their results. Its popularization and further development have been spearheaded by the interacting MM (IMM) algorithm, which was proposed by Bar-Shalom and Blom [3]. The performance of the IMM algorithm suffers when there are too many motion models that overlap and compete with each other [4–6]. To solve this problem, X. Rong Li proposed the third-generation MM algorithm, i.e., the variable-structure multiple-model (VSMM) algorithm [7]. The model set within this model can be adaptively adjusted according to the changes in the motion mode of the target [8–10].

The target motion mode may change to avoid obstacles. For example, an automated guided vehicle (AGV) will change the way forward to bypass buildings and objects on the ground; a general aviation aircraft will change direction to avoid no-fly zones, bad meteorological zones along a route, etc. For the type of maneuvering that is performed in order to avoid all obstacles, existing research

shows that a combination of the obstacle information and the MM algorithm can achieve a better tracking performance [11–13].

According to the description of the obstacles, incorporating the available obstacle information into the MM algorithms can be divided into two categories: implicit and explicit. In the implicit category, instead of obstacles, a permitted range of motion is given. For example, road information is used to adjust the model set in References [13–15], in which areas that are off of the road are considered to be obstacles. In addition to model set adjustment, a modification of the model probability (MP) and the model transition probability matrix (TPM) are presented in references [16] and [17], where guard conditions on waypoints that must be flown over, are used to alter the update of the MP and/or the TPM. In the explicit category, the range of obstacles is given directly. For example, circular obstacles are presented in the state-dependent variations of the interacting multiple-model (SD-IMM) algorithm by Rastgoufard [11]. The SD-IMM algorithm uses the distances between the target position and the obstacles as auxiliary information to modify the MP in the updating step and the TPM in the mixing step of the IMM algorithm. Its performance is always clearly better than that of the traditional version [12].

Using circles to describe obstacles is attractive because of its simplicity in computing, but it may mistake non-obstacle areas as obstacle areas. Furthermore, the SD-IMM algorithm gives the same adjustment value for the same distance between a target and obstacle circles with different radiuses. However, for obstacles with different sizes, evasive maneuvering should be different, since the target may require slighter maneuvers to bypass small obstacle circles than it does to bypass large obstacle circles.

To solve the problems mentioned above, we use polygons to describe the obstacles and introduce a new piece of obstacle information, i.e., the field angle. Fuzzy inference systems are applied to simplify the complex relationship between the obstacle information and the update of the MP and TPM; thus, a fuzzy-logic-based, obstacle information-aided multiple-model (FOIA-MM) algorithm is presented.

The rest of this paper is structured as follows: a brief introduction to the stochastic model and the improvement of the SD-IMM algorithm is presented in Section 2. In Section 3, the obstacle information description is given first, followed by the method to adjust MP and TPM, through their utilization. In Section 4, the performance of the proposed algorithm is compared with that of the SD-IMM algorithm by simulation and is illustrated by a UAV experiment, the results of which show that the proposed algorithm is effective and efficient. Finally, the conclusions are presented in Section 5.

## 2. Outline of the SD-IMM Algorithm

This section presents the stochastic model and the improvement of the SD-IMM algorithm.

### 2.1. Stochastic Model

In the SD-IMM algorithm, the maneuvering target is typically modeled through "hybrid systems", which means the target state is a continuous process, while its motion modes are described as a finite model set. Assuming that there are $r$ models matching the motion mode currently in effect, $M_k = \{m_k^1, m_k^2, m_k^3, \ldots m_k^r\}$, the dynamics equation and the measurement equation are, respectively, defined by:

$$x_{k+1} = F_k^i(m_k^i)x_k + w_k^i(m_k^i) \tag{1}$$

$$z_k = H_k^i(m_k^i)x_k + v_k^i(m_k^i) \tag{2}$$

where $x$ denotes an $n$-dimensional state vector, and $z$ denotes an $m$-dimensional measurement vector; $F_k^i(\cdot)$ and $H_k^i(\cdot)$ are the state transition function and the measurement function, respectively; $i \in \{1, 2, \cdots, r\}$, $m_k^i$ is the $i$-th adopted model at time $k$. The random variables $w_k^i$ and $v_k^i$ represent the process noise and the measurement noise, respectively. They are mutually independent zero-mean Gaussian white noise with covariance $\text{cov}(w_k^i) = Q_k^i$ and $\text{cov}(v_k^i) = R_k^i$.

The transition between different models is typically regarded to as Markov chain. The model transition probability $p_{ji}$ is given by:

$$p_{ji} = P(m_k^i | m_{k-1}^j) \tag{3}$$

where $m_{k-1}^j$ is the $j$-th adopted model at time $k-1$.

*2.2. Improvement in the SD-IMM Algorithm*

The SD-IMM algorithm improves the IMM algorithm by applying the state-dependent obstacle information to adjust the MP and TPM. Suppose that there are $P$ circular obstacles, each with the radius $r_l$ and the center $x_l, y_l, l \in \{1, 2, \cdots, P\}$; $d_l(x)$, the distance between the position of the state $x$ and the $i$-th obstacle, can be calculated as:

$$d_l(x) = \sqrt{(x - x_l)^2 + (y - y_l)^2} - r_l \tag{4}$$

where $(x, y)$ is the position of the elements in $x$.

The influence level of the $l$-th obstacle circle on the state $x$ is defined as:

$$S(x, l) = \frac{1}{1 + e^{-\beta(d_l(x))}} \tag{5}$$

where $\beta$ is the shape of the parameter. Thus, the obstacle information, $S(x)$, can be obtained by:

$$S(x) = \min_{1 \le l \le p} S(x, l) \tag{6}$$

Unlike the traditional IMM, the SD-IMM algorithm adjusts the update of MP $\mu_k^i$ and TPM $= [p_{ji,k}]$ as follows:

$$\mu_k^i = \frac{\mu_{k|k-1}^i \Lambda_k^i s_k^i}{\sum_j \mu_{k|k-1}^j \Lambda_k^j s_k^j}, \quad s_k^i = S(\hat{x}_{k|k-1}^i) \tag{7}$$

$$p_{ji,k} = \frac{p_{ji,k-1} s_{k|k-1}^{ji}}{\sum_j p_{ji,k-1} s_{k|k-1}^{ji}}, \quad s_{k|k-1}^{ji} = S(\hat{x}_{k|k-1}^{ji}) \tag{8}$$

where $\Lambda_k^i$ is the likelihood of model $m_k^i$; $\hat{x}_{k|k-1}^i$ is the predicted state of model $m_k^i$; $\hat{x}_{k|k-1}^{ji}$ is the updated state of model $m_k^i$. This adjustment incorporates the obstacle information into the IMM algorithm and thus makes the SD-IMM algorithm achieve a better tracking performance [12].

## 3. FOIA-MM Algorithm

Although the SD-IMM algorithm outperforms the traditional IMM algorithm in the obstacle scene, it has two main shortages: one is the mistake of viewing the non-obstacle areas as obstacle areas by using circles to describe them. As shown in Figure 1, the shaded parts are non-obstacle areas that are mistaken as obstacle areas. The crosses represent the true positions of the target, while the small circles show the estimates made by the SD-IMM algorithm. It shows that the SD-IMM algorithm makes several wrong estimates for taking a triangular obstacle as a circular obstacle. This shortage may lead to large errors in the estimation or to unnecessary maneuvers in the navigation. The other shortage is the insufficiency of utilizing only the distance between a target and obstacle circles in order to adjust the MP and TPM. Since a target's evasive maneuvering for avoiding small obstacles must be slighter than that for large obstacles, the target needs a big turn rate to bypass a large circular obstacle and a little turn rate to bypass a small circular obstacle. As shown in Figure 2, *Target* (1) takes a greater turn rate than *Target* (2) ($\beta(1) > (\beta(2))$) to bypass obstacles of different sizes when they are at the same

distance from the obstacles. The angles $\beta(1)$ and $\beta(2)$ are defined as the field angles, which are relative to *Area* (1) and *Area* (2).

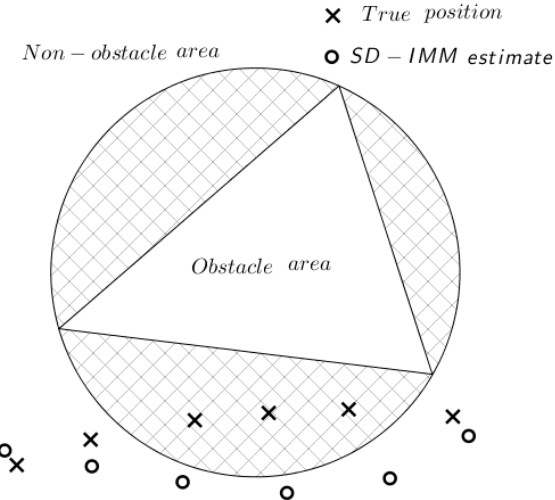

**Figure 1.** State-dependent variations of the interacting multiple-model (SD-IMM) algorithm scenario, with estimates at the obstacle area (three error estimates).

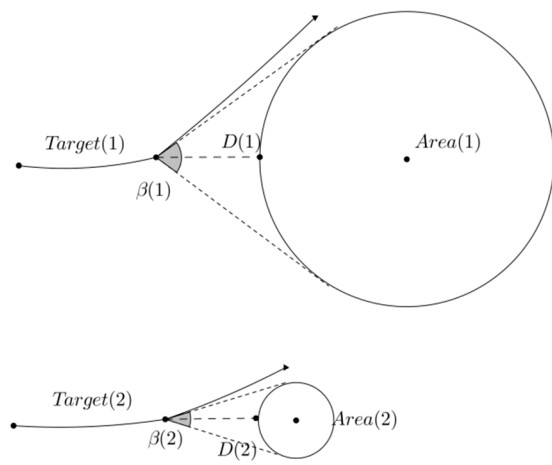

**Figure 2.** Target's evasive maneuvering for different-size obstacles (D(1) = D(2), β(1) > β(2)).

To overcome the above two shortages, we applied polygons to describe the obstacles and used a new metric, the field angle (the angular extent of the obstacle viewing at target position), which was designed to reflect the size of the obstacles. Both the distance and the field angle were used to adjust the MP and the TPM in the proposed algorithm. For targets at the same distance to obstacles with different sizes, the bigger the field angle, the stronger the target maneuvers, and vice versa. It is hard to express this qualitative relationship mathematically, so fuzzy inference was employed. The fuzzy inference has been successfully applied to target tracking problems and has achieved good performance [18,19]. On the basis of the improvements mentioned above, we propose the fuzzy-logic-based obstacle information-aided multiple-model (FOIA-MM) algorithm.

Figure 3 shows the schematic block diagram of the proposed algorithm. First, the obstacle information, i.e., the distance $S_k$ and the field angle $B_k$, are calculated according to the predicted target position. Next, the obstacle information is fed into two fuzzy logic inference systems: one system outputs the model weights to their probabilities, the other yields the expected sojourn time of the models for the assignment of TPM. At last, the target state estimation is obtained through MM filtering. More details can be found in the following section.

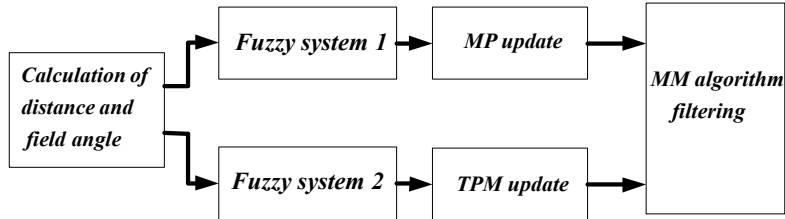

**Figure 3.** The schematic block diagram of the fuzzy-logic-based obstacle information-aided multiple-model (FOIA-MM) algorithm.

### 3.1. Obstacle Information Descriptions

In practical applications, many obstacles have various shapes. However, in order to avoid a too large computational load, the FOIA-MM algorithm simplifies the shape of the obstacle into a polygonal shape. A concave polygon will produce abrupt changes in practical applications, which will greatly affect the tracking performance and even reduce the tracking performance. If it is a concave polygon, it must first be filled as a convex polygon. The convex polygon used in this algorithm is enough to cover the shape of obstacles. To simplify, for obstacles with a non-linear shape, this study used a convex polygon, which can best cover their shape. The FOIA-MM algorithm will declare all polygons as convex polygons.

Suppose there are $P$ obstacle convex polygons in the motion scene. The predicted target position is $x_{k|k-1}$. The distance $D(j)$ from $x_{k|k-1}$ to the $j$-th obstacle *Area(j)* is calculated as follows:

$$D(j) = \inf_{y \in Area(j)} \|x_{k|k-1} - y\| \tag{9}$$

where $\| \ \|$ is the Euclidean norm. The field angle, $\beta(j)$, relative to $D(j)$, is half the sum of the angular extent of all the sides of *Area(j)* viewing at $x_{k|k-1}$, $\beta(j) = \frac{1}{2}\sum_i \beta_{ji}$ ($\beta_{ji}$ is the angular extent of the $i$-th side of Area(j)). The following rules can be obtained according to geometric knowledge. An example of the scene is shown in Figure 4. It has three obstacle areas; the dark point $T$ is $x_{k|k-1}$. $\beta(3)$ is the biggest field angle between $x_{k|k-1}$ and the three obstacle areas.

$$\begin{cases} \beta(j) = \pi, & x_{k|k-1} \text{ is inside the obstacle area} \\ \beta(j) < \pi, & x_{k|k-1} \text{is outside the obtascle area} \end{cases} \tag{10}$$

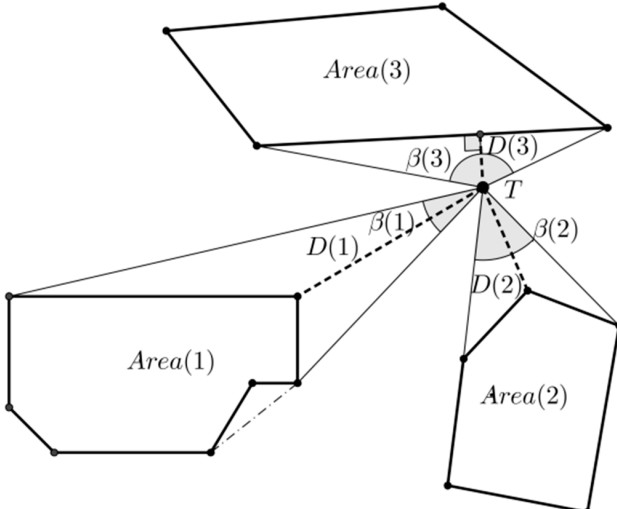

**Figure 4.** Distance between x_(k|k − 1) and obstacle convex polygons (D(3) < (D(1), D (2))); field angle between $x_{k|k-1}$ and obstacle convex polygons ($\beta(3) > (\beta(1), \beta(2))$).

As the target moves far away from the obstacle area, $\beta(j)$ approaches zero. Based on the distance $D(j)$ and the field angle $\beta(j)$, the obstacle information $B_k$ and $S_k$ can be calculated as follows:

$$B_k = \beta(j*) \quad S_k = D(j*) \tag{11}$$

where

$$j* = \arg \min_{1 \leq j \leq P} D(j) \tag{12}$$

An example of the scene is shown in Figure 4. It has three obstacle areas; the dark point $T$ is $x_{k|k-1}$. $D(3)$ is the shortest distance between $x_{k|k-1}$ and three obstacle areas. Therefore, the obstacle information is $S_k = D(3)$ and $B_k = \beta(3)$.

### 3.2. MP Update

Suppose $J_k^i$ is the weight of model $m_k^i$, which reflects the influence of the obstacles on the MP. The weighted model probability of $\mu_k^i$ is [11]:

$$\mu_k^i = \frac{1}{C} \Lambda_k^i J_k^i \sum_{j=1}^{r} p_{ji} \mu_{k-1}^j \tag{13}$$

where C is a normalizing factor, and $\mu_{k-1}^j$ is the model probability at the time $k - 1$.

The relationship between $J_k^i$ and the obstacle information $(S_k, B_k)$ is not easy to quantify, so $J_k^i$ is acquired by using fuzzy inference in the proposed algorithm. The fuzzy inference system has two inputs, $S_k$ and $B_k$, and one output, $J_k^i$. The universe of discourse of $S_k$ is mapped into three fuzzy sets: near (NE), medium (ME), and far (FA), as shown in Figure 5. That of $B_k$ is also divided into three fuzzy sets: small positive (SP), medium positive ($\overline{\text{MP}}$), and large positive (LP), as shown in Figure 6. The universe of discourse of $J_k^i$ is partitioned into five fuzzy sets, labeled in the linguistic terms of zero (ZE), low (LOW), medium (ME), high (HI), and very high (VH), as shown in Figure 7.

The fuzzy inference rules for $J_k^i$ are described as follows. When $x_{k|k-1}$ is near the obstacle areas, $S_k$ is small and $J_k^i$ should be small. When $x_{k|k-1}$ is not near the obstacle areas, if $S_k$ increases, $J_k^i$ should increase. If $B_k$ increases, which means that no matter how big an obstacle is, target tends to require higher maneuvers to avoid hitting the target, then $J_k^i$ should decrease.

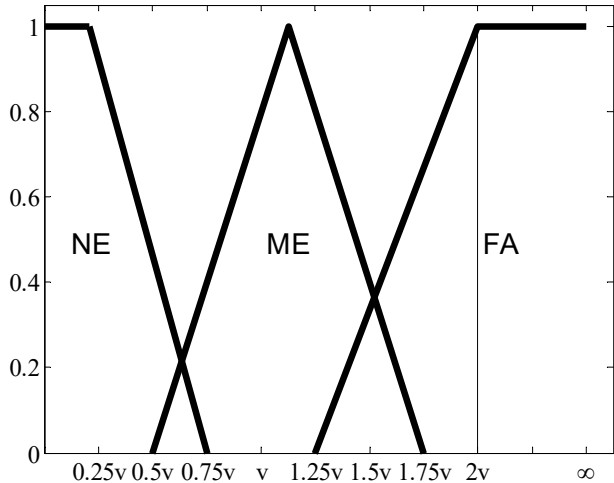

**Figure 5.** Membership functions of fuzzy sets in distance $S_k$ ($v$ is the current target velocity).

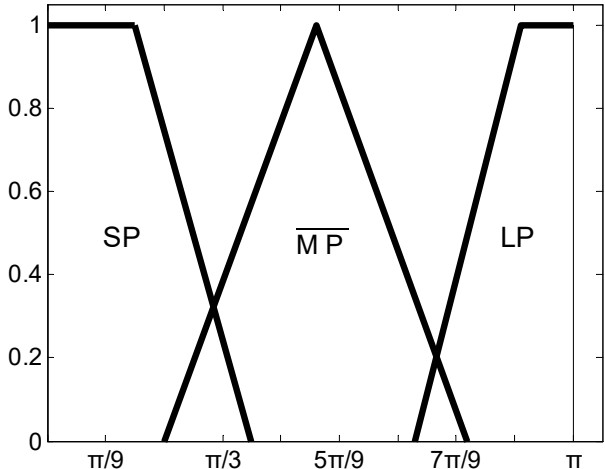

**Figure 6.** Membership functions of fuzzy sets in field angle $B_k$.

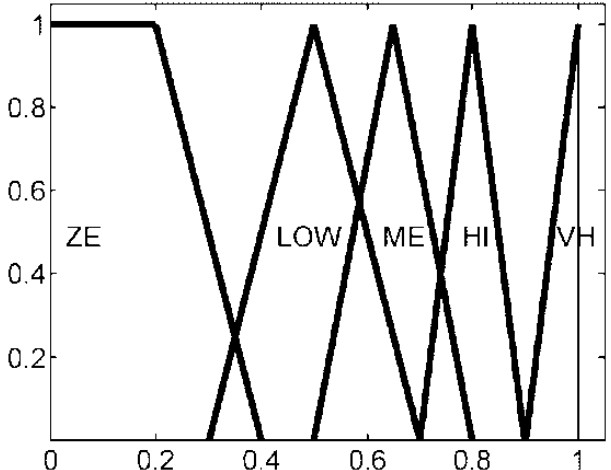

**Figure 7.** Membership functions of fuzzy sets in weight $J_k^i$.

### 3.3. TPM Update

The performance of the MM algorithm is related to the choice of the TPM [20], which is easily affected by the obstacle areas. Let $\tau_k^i$ be the expected sojourn time of the *i*-th model. The diagonal elements $p_{ii}$ of the TPM are given by [21]:

$$
\begin{aligned}
p_{ii} &= 1 - \frac{SI}{\tau_k^i} \quad \tau_k^i \geq SI \\
&i \in \{1, 2, \cdots, r\}
\end{aligned}
\tag{14}
$$

where *SI* is the sampling interval. After $p_{ii}$ has been determined, the off-diagonal elements $p_{ji}$ of the TPM are given by:

$$
\begin{aligned}
p_{ji} &= \frac{1-p_{ii}}{r-1} \\
i,j &\in \{1, 2, \cdots, r\}
\end{aligned}
\tag{15}
$$

The expected sojourn time, $\tau_k^i$, of a model is the amount of time the model is expected not to change. It is apparently affected by obstacles or by $(S_k, B_k)$. When $x_{k|k-1}$ is near the obstacle areas, $S_k$ is small, and $\tau_k^i$ should be small. When the target moves away from the obstacle areas, if $S_k$ increases, $\tau_k^i$ should also increase. If $B_k$ increases, which means that no matter how big an obstacle is, target tends to require higher maneuvers to avoid hitting the target, then $\tau_k^i$ should decrease.

According to the relationship between $\tau_k^i$ and $(S_k, B_k)$ stated above, $\tau_k^i$ is acquired by using fuzzy inference in the proposed algorithm. For both $S_k$ and $B_k$, the universe of discourse is mapped into three

fuzzy sets, as shown in Figures 5 and 6, respectively. The acquired $\tau_k^i$ is partitioned into five fuzzy sets, labeled in the linguistic terms of very short (VS), short (SH), medium (ME), long (LO), and very long (VL), as shown in Figure 8.

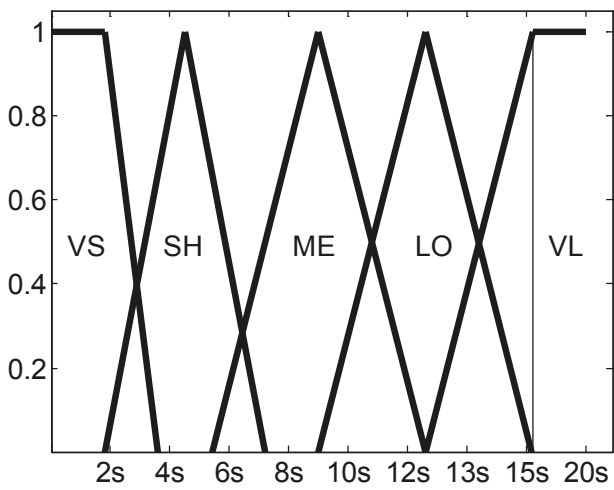

**Figure 8.** Membership functions of fuzzy sets in $\tau_k^i$ (expected sojourn time of the *i*-th model).

### 3.4. The Iterative Process of the FOIA-MM Algorithm

The FOIA-MM algorithm combines the obstacle information into the MM algorithm. The iterative process of the FOIA-MM algorithm is given in Table 1. Filters in it are Kalman filters.

**Table 1.** The iterative process of the FOIA-MM algorithm.

---

**1. Conditioned (re)initialization**

Predicted MP: $\mu_{k|k-1}^i = p\left\{m_k^i \middle| M_k, M_{k-1}, Z^{k-1}\right\} = \sum p_{ji}\mu_{k-1}^j$

Mixing MP: $\mu_{k-1}^{j|i} = p\left\{m_{k-1}^j \middle| m_k^i, M_{k-1}, Z^{k-1}\right\} = p_{ji}\mu_{k-1}^j / \mu_{k|k-1}^i$

Mixing state: $\bar{x}_{k-1}^i = E[x_{k-1}|m_k^i, Z^{k-1}] = \sum \hat{x}_{k-1|k-1}^j \mu_{k-1}^{j|i}$

Mixing covariance: $\overline{P}_{k-1}^i = \sum \left\{P_{k-1|k-1}^j + [\bar{x}_{k-1}^i - \hat{x}_{k-1|k-1}^j][\bar{x}_{k-1}^i - \hat{x}_{k-1|k-1}^j]^T\right\}\mu_{k-1}^{j|i}$

---

**2. Kalman filtering**

Predicted state: $\hat{x}_{k|k-1}^i = E(x_k|m_k^i, M_{k-1}, Z^{k-1}) = F_{k-1}^i \bar{x}_{k-1}^i$

Predicted covariance: $P_{k|k-1}^i = F_{k-1}^i \overline{P}_{k-1}^i (F_{k-1}^i)^T + Q_{k-1}^i$

Measurement residual: $\bar{z}_k^i = z_k - E(z_k|m_k^i, M_{k-1}, Z^{k-1}) = z_k - H_k^i \hat{x}_{k|k-1}^i - \bar{v}_k^i$

Residual covariance: $S_k^i = H_k^i P_{k|k-1}^i (H_k^i)^T + R_k^i$

Filter gain: $K_k^i = P_{k|k-1}^i (H_k^i)^T (S_k^i)^{-1}$

Updated state: $\hat{x}_{k|k}^i = E(x_k|m_k^i, M_{k-1}, Z^{k-1}) = \hat{x}_{k|k-1}^i + K_k^i \bar{z}_k^i$

Updated covariance: $P_{k|k}^i = P_{k|k-1}^i - K_k^i S_k^i (K_k^i)^T$

---

**3. updated MP and TPM**

Model likelihood: $\Lambda_k^i = p(\bar{z}^i|m_k^i, M_{k-1}, Z^{k-1}) \overset{assume}{=} N[\bar{z}_k^i; 0, S_k^i]$

Model weight: $B_k, S_k \overset{Fuzzy}{\rightarrow} J_k^i$

Updated MP: $\mu_k^i = \frac{1}{C}\Lambda_k^i J_k^i \sum_i p_{ji}\mu_{k-1}^j$

Expected sojourn time: $B_k, S_k \overset{Fuzzy}{\rightarrow} \tau_k^i$

Updated TPM: $p_{ii} = 1 - \frac{T}{\tau_k^i} \quad p_{ij} = \frac{1-p_{ij}}{r-1}$

---

**4. Combination**

Overall state: $\hat{x}_{k|k} = E(x_k|M_k, M_{k-1}, Z^k) = \sum \hat{x}_{k|k}^i \mu_k^i$

Overall covariance: $P_{k|k} = \sum [P_{k|k}^i + (\hat{x}_{k|k} - \hat{x}_{k|k}^i)(\hat{x}_{k|k} - \hat{x}_{k|k}^i)^T]\mu_k^i$

---

## 4. Experimental Results and Analysis

In this section, simulation experiments were carried out to compare the performances of the FOIA-MM algorithm and the SD-IMM algorithm. A UAV experiment was also carried out to demonstrate the effectiveness of the proposed algorithm in real applications.

### 4.1. Simulation Scenario

The position and velocity were chosen as state elements in the target-tracking simulation. That is:

$$x = [x, \dot{x}, y, \dot{y}] \tag{16}$$

where $x, y$ are the target coordinates, and $\dot{x}, \dot{y}$ are the velocities. The scenario was designed as shown in Figure 9. Three obstacles are labeled *Area* (1), *Area* (2), and *Area* (3). The thick black line indicates the boundary of the obstacle areas, while the thin black line shows the true target trajectory. *Area* (3) is relatively small with regard to *Area* (1) and *Area* (2). *Position (1)* and *Position (2)* have the same distance from the nearest obstacle *Area* (2) and *Area* (3). *Position (1)* has a larger field angle than *Position* (2), which leads to a larger turning maneuver to avoid hitting the obstacle. The initial state of the target is given by $x_0$ = (180 m, 0 m/s, 49 m, 30 m/s). The target trajectory has nine phases, as shown in Table 2. These three obstacles were simplified into six circular obstacle areas in the SD-IMM algorithm, as shown in Figure 10, each with radius $r_l$ and center $x_l, y_l$, $l \in \{1, 2, \cdots, 6\}$. In this paper, the FOIA-MM algorithm simplified the obstacle into a convex polygon, and the SD-IMM algorithm simplified the obstacle into a circle. If an obstacle is covered with multiple circles, the number of circles needed is relatively large, which will greatly increase the computational complexity of the multiple-model algorithm with heavy computational load, and the real-time performance of the algorithm is difficult to be guaranteed. The target trajectory crossed over the circular obstacle areas several times. In the SD-IMM algorithm, when the target state estimation falls into the circular obstacle area, the weight of a model reaches the minimum.

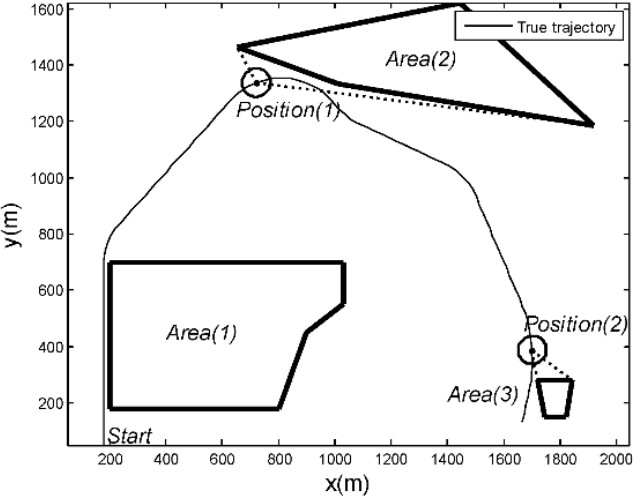

**Figure 9.** The target's true trajectory and motion scenario.

**Table 2.** Motion models and durations of the target's trajectory.

| Model | CV | CT $+ 7°/s$ | CV | CT $+ 7°/s$ | CT $- 5°/s$ |
|---|---|---|---|---|---|
| Time (s) | $1 \sim 22$ | $23 \sim 27$ | $28 \sim 46$ | $47 \sim 60$ | $61 \sim 67$ |

| Model | CT $+ 7°/s$ | CV | CT $+ 3°/s$ | CV |
|---|---|---|---|---|
| Time (s) | $68 \sim 75$ | $76 \sim 85$ | $86 \sim 95$ | $96 \sim 110$ |

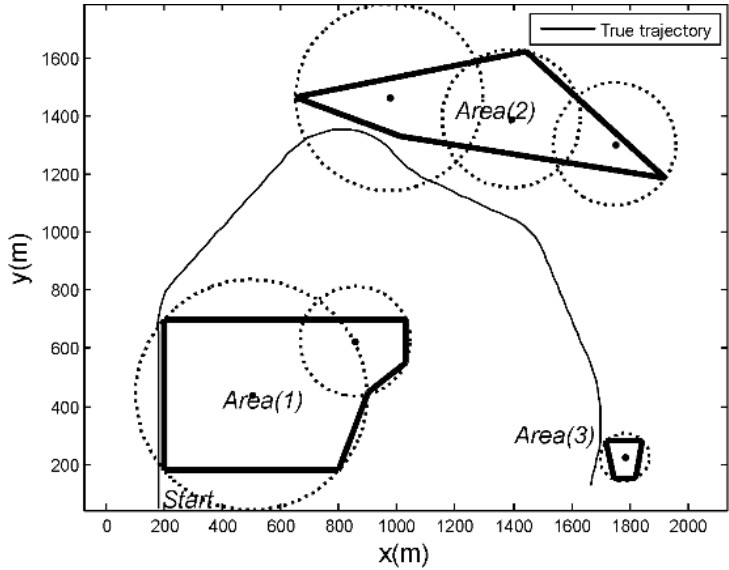

**Figure 10.** Scenario of the SD-IMM algorithm.

The initial model set of the proposed algorithm and the SD-IMM algorithm consists of five models: a constant-velocity (CV) model and four coordinated-turn (CT) models with disparate turn rates: $\omega_1 = +3°/s, \omega_2 = +5°/s, \omega_3 = +7°/s, \omega_4 = -5°/s$. The initial values of MP were [0.2, 0.2, 0.2, 0.2, 0.2]. The dynamics matrix of the CT model is expressed by:

$$F_k^i = \begin{bmatrix} 1 & \frac{\sin \omega_i T}{\omega_i} & 0 & -\frac{1-\cos \omega_i T}{\omega_i} \\ 0 & \cos \omega_i T & 0 & -\sin \omega_i T \\ 0 & \frac{1-\cos \omega_i T}{\omega_i} & 1 & \frac{\sin \omega_i T}{\omega_i} \\ 0 & \sin \omega_i T & 0 & \cos \omega_i T \end{bmatrix}$$
$$i = 1, 2, 3, 4 \cdots$$

(17)

The initial values of TPM were

$$\begin{bmatrix} 0.9 & 0.025 & 0.025 & 0.025 & 0.025 \\ 0.05 & 0.8 & 0.05 & 0.05 & 0.05 \\ 0.05 & 0.05 & 0.8 & 0.05 & 0.05 \\ 0.05 & 0.05 & 0.05 & 0.8 & 0.05 \\ 0.05 & 0.05 & 0.05 & 0.05 & 0.8 \end{bmatrix}$$

(18)

The measurements included the zero-mean Gaussian noises with standard deviations of $\sigma_1 = 30m$ and $\sigma_2 = 40m$ in the simulation trajectory. Tables 3 and 4 are the fuzzy rules used in the simulation.

**Table 3.** Fuzzy rules for $S_k$, $B_k$ and $J_k^i$.

| $J_k^i$ | | $S_k$ | | |
|---|---|---|---|---|
| | | **VN** | ME | FA |
| $B_k$ | ZE | ZE | LOW | HI |
| | $\overline{MP}$ | ZE | ME | HI |
| | LP | ZE | LOW | LOW |

**Table 4.** Fuzzy rules for $S_k$, $B_k$ and $\tau_k^i$.

| $\tau_k^i$ | | $S_k$ | | |
|---|---|---|---|---|
| | | **VN** | ME | FA |
| $B_k$ | ZE | VS | ME | VL |
| | $\overline{\text{MP}}$ | VS | LO | VL |
| | LP | VS | VS | SH |

*4.2. Performance Comparison*

To evaluate the performance of the FOIA-MM algorithm, the root-mean-square error (RMSE) in the position and velocity was used. The results are the average of 50 Monte Carlo simulations. Figures 11 and 12 show the RMSE in position and velocity estimation, with measurement noise standard deviations of $\sigma_1 = 30$ and $\sigma_2 = 40m$, respectively. Meanwhile, Figures 11 and 12 also show the distribution of the distance between the target's true positions and the obstacle areas. They demonstrate that most RMSE of the estimation of the SD-IMM algorithm were larger than those of the proposed algorithm, especially when the target was close to the edge of the obstacle areas. When the measurement noise level was bigger, the performance of the proposed algorithm was better than that of the SD-IMM algorithm. The MP at *Position* (1) and *Position* (2) are shown in Table 5. The real-motion model at *Position* (1) is $CT + 7°/s$, and that at *Position* (2) is $CT + 3°/s$. As shown in Table 5, the FOIA-MM algorithm gave a more correct probability of the real models than the SD-IMM algorithm. This validated the effect of the field angle. In addition, as shown in Table 6, the proposed algorithm decreased the extent of the wrong estimation compared with the SD-IMM algorithm through statistical analysis. This result is the sum of 50 tests.

Computational complexity is an important index that measures whether an algorithm has real-time processing capability, which is especially important in tracking systems with high real-time requirements. In the field of tracking, many algorithms with high tracking accuracy are limited because of their high computational complexity. In this paper, the parameters of the simulation platform are: CPU Intel i3 2.0 GHz, 4 GB of memory, and operating environment MATLAB R2009b, which can perform 100 Monte Carlo simulations. The average CPU running time of FOIA-MM algorithm and SD-IMM algorithm was 0.234 s and 0.181 s, respectively. Because fuzzy inference requires a certain amount of time, the FOIA-MM algorithm ran longer than the SD-IMM algorithm.

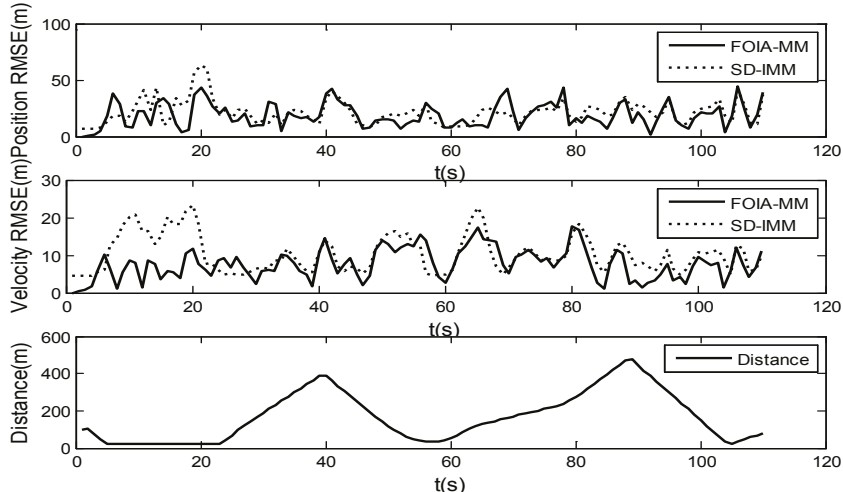

**Figure 11.** Root-mean-square error (RMSE) of position (m) and velocity (m/s); ($\sigma_1 = 30m$), distance between the true position and the obstacle areas.

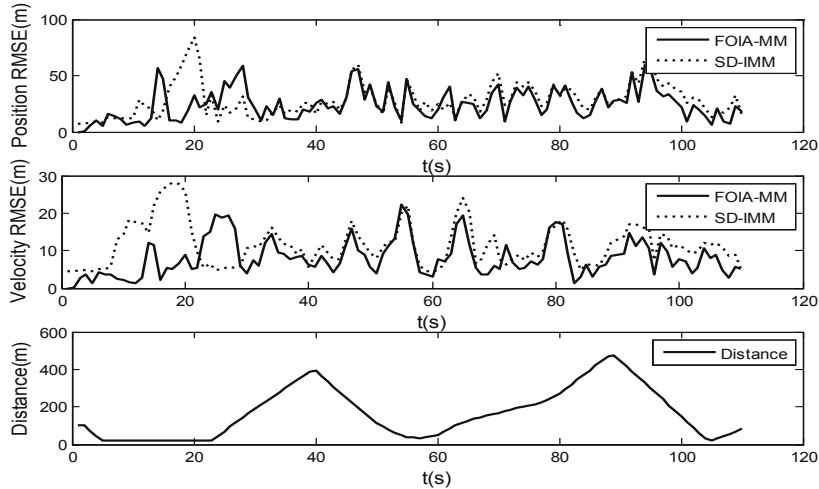

**Figure 12.** RMSE of position (m) and velocity (m/s); ($\sigma_2 = 40m$), distance between the true position and the obstacle areas.

**Table 5.** Model probabilities of Position (1) and Position (2).

|  | Model | CV | $CT + 3°/s$ | $CT + 5°/s$ | $CT - 5°/s$ | $\mathbf{CT+7°/s}$ |
|---|---|---|---|---|---|---|
| *Position (1)* | FOIA-MM | 0.09 | 0.13 | 0.11 | 0.04 | **0.63** |
| | SD-IMM | 0.14 | 0.16 | 0.20 | 0.14 | **0.36** |
|  | Model | CV | $\mathbf{CT + 3°/s}$ | $CT + 5°/s$ | $CT - 5°/s$ | $CT + 7°/s$ |
| *Position (2)* | FOIA-MM | 0.15 | **0.60** | 0.16 | 0.05 | 0.04 |
| | SD-IMM | 0.21 | **0.25** | 0.24 | 0.10 | 0.20 |

The dark columns are real-model and the estimated-model probabilities by FOIA-MM and SD-IM. The table shows that FOIA-MM gave higher correct model probabilities than SD-IMM.

**Table 6.** Number of positions located in the obstacle areas.

|  | Measured | Estimated | Improvement |
|---|---|---|---|
| FOIA-MM | 655 | 5 | 99.24% |
| SD-IMM | 655 | 490 | 25.2% |

## 4.3. Field Experiment and Results Analysis

In this experiment, the performance of the FOIA-MM algorithm was verified for tracking a UAV in a real scenario, as shown in Figure 13. The thick black line indicates the boundary of two high-rise buildings. The black asterisks indicate measurements collected by the GPS sensor during the flight of the UAV. The thin black line shows the FOIA-MM algorithm filtering results. About 10% measurements fell in the obstacle areas due to the GPS system measurement error. They were wrong measurements. Figure 13 shows that the FOIA-MM algorithm could successfully track the UAV and filter out all the wrong measurements.

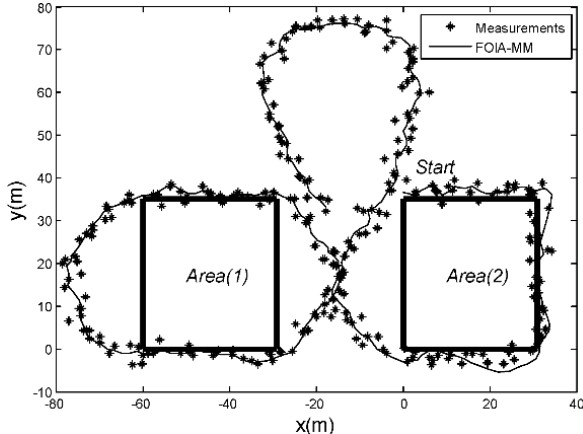

**Figure 13.** The filtering results of the UAV experiment.

## 5. Conclusions

In this paper, we presented the FOIA-MM algorithm for incorporating the obstacle information to the multiple model algorithm. We used convex polygons to describe obstacles to avoid mistaking non-obstacle areas as obstacle areas. For obstacles with different sizes, evasive maneuvering is different. We introduced a new term, the field angle, in the obstacle information, which can be considered as a relative distance between the target and the obstacles. The proposed algorithm regards the distance and the field angle as the obstacle information and analyses this relationship by mapping them to a set of fuzzy rules, which will modify the TPM and the MP for the state estimate. The effectiveness is evaluated by simulation experiments and a UAV experiment. These experiments have shown that the proposed algorithm has a good performance when compared to the SD-IMM algorithm. The fuzzy rules used in this paper can improve the performance currently achieved through their adjustment. Fuzzy rules are difficult to adjust, and erroneous rules will hamper improving the tracking performance. How to determine appropriate fuzzy inference rules is a critical problem for future works. The proposed algorithm can be extended to solve robot localization problems, drone tracking and localization, and other practical application problems.

**Author Contributions:** Conceptualization, Q.W.; methodology, Q.W.; validation, Q.W.; formal analysis, E.F.; investigation, P.L.; data curation, Q.W.; writing—original draft preparation, Q.W.; writing—review and editing, Q.W.; supervision, Q.W.; project administration, Q.W.

**Funding:** This work was support by the National Natural Science Foundation of China: 61703280, the National Natural Science Foundation of China: 61603258, the Plan Project of Science and Technology of Shaoxing City: 2017B70056; Military Postgraduate Foundation in the Whole Army: Study on aerial information application of air defense multi-source radar network.

**Conflicts of Interest:** The authors declared that they have no conflicts of interest to this work. We declare that we do not have any commercial or associative interest that represents a conflict of interest in connection with the work submitted.

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
