# Peer review of "Fuzzy-Logic-Based, Obstacle Information-Aided Multiple-Model Target Tracking"

_information, doi:10.3390/info10020048_

Reviewer 1 Report

This paper presents a fuzzy-logic-based method of incorporating new obstacle information into the IMM algorithm.

The reviewer has several comments.

The name of the paper does not present that the paper is on target tracking.

I think the following name is better: "fuzzy-logic-based obstacle information aided multiple model target tracking".

At line 137 of page 5, beta_{ji} is introduced. beta_{ji} is defined as "the angular extent of the i-th side of Area(j)". I think the definition of beta_{ji} is not clear.

(10) only holds in the case where all obstacles are convex. Do you consider convex obstacles?

If so, please clarify that this paper considers convex obstacles.

At line 188 of page 8, the following sentence does not make sense:

"algorithm is given in..."

In simulation section, which algorithm did you use to derive polygon from arbitrarily shaped obstacle?

Also, which algorithm did you use to derive circles from arbitrarily shaped obstacle?

Fig 10 shows that three circles are used to cover an obstacle.

However, I think you can pack many circles inside the obstacle so that they cover the obstacle more tightly.

Also, the triangle in Fig 1 can be covered by many small circles so that they cover the obstacle more tightly.

Computational load of your algorithm is not presented.

Also, computational load of SD-IMM is not presented.

Author Response

Response to Reviewer 1 Comments

Point 1: The name of the paper does not present that the paper is on target tracking. I think the following name is better: "fuzzy-logic-based obstacle information aided multiple model target tracking".

Response 1: 

I think the comments are very good. I have changed the title of the paper to "fuzzy-logic-based obstacle information aided multiple model target tracking".

Corresponding modification:

At line 1of page 1.

Point 2: At line 137 of page 5, beta_{ji} is introduced. beta_{ji} is defined as "the angular extent of the i-th side of Area(j)". I think the definition of beta_{ji} is not clear.

Response 2:

 I have modified this  paper in several places to make the field angle beta clearer.

Corresponding modification:

At line 114-115 of page 4.

At line 154-155of page 5.

Point 3: (10) only holds in the case where all obstacles are convex. Do you consider convex obstacles? If so, please clarify that this paper considers convex obstacles.

Response 3:

In practical application, many obstacles have various shapes. However, in order not to add too much computational load, FOIA-MM algorithm simplified the shape of the obstacle into a polygonal shape. Concave polygon will produce abrupt changes in practical application, which will greatly affect the tracking performance and even reduce the tracking performance. If it is concave polygon, it must first be filled as convex polygon. Convex polygon used in this algorithm is enough to cover the shape of obstacles. For obstacles with non-linear shape, This paper use convex polygon, which can best cover its shape, to simplify. FOIA-MM algorithm will declare all polygon as convex polygons.

Corresponding modification:

At line 138-145 of page 5.

Point 4: In simulation section, which algorithm did you use to derive polygon from arbitrarily shaped obstacle? Also, which algorithm did you use to derive circles from arbitrarily shaped obstacle? Fig 10 shows that three circles are used to cover an obstacle. However, I think you can pack many circles inside the obstacle so that they cover the obstacle more tightly. Also, the triangle in Fig 1 can be covered by many small circles so that they cover the obstacle more tightly.

Response 4:

In this paper, FOIA-MM algorithm simplified the obstacle into convex polygon, and SD-IMM algorithm simplified the obstacle into a circle. If I cover an obstacle with multiple circles, the number of circles needed is relatively large, which will greatly increase the computational complexity of the multiple model algorithm with heavy computational load, and the real-time performance of the algorithm is difficult to be guaranteed.

Corresponding modification:

At line 222-226 of page 9.

Point 5: Computational load of your algorithm is not presented. Also, computational load of SD-IMM is not presented.

Response 5:

I have added the analysis part of computational complexity in this article.  Computational complexity is an important index to measure whether an algorithm has real-time processing capability, which is especially important in tracking systems with high real-time requirements. In the field of tracking, many algorithms with high tracking accuracy are limited due to their high computational complexity. In this paper, the parameters of the simulation platform are: CPU Intel i3 2.0 GHz, 4 GB of memory, and the operating environment MATLAB R2009b, which can perform 100 Monte Carlo simulations.  The average CPU running time of FOIA-MM algorithm and SD-IMM algorithm is 0.234 seconds and 0.181 seconds respectively. Because fuzzy inference requires a certain amount of time, the FOIA-MM algorithm runs longer than the SD-IMM algorithm.

Corresponding modification:

At line 263-271 of page 11.

Reviewer 2 Report

This paper utilizes the distance and the field angle as the obstacle information and analyzes their relationship, and then maps them to a set of fuzzy rules, which will modify the TPM and MP for the state estimate. In my opinion, this topic is interesting. Nevertheless, addressing the following issues will improve the paper further.

1. The proposed algorithm uses polygons to describe the obstacles and then extracts the distance and the field angle of these obstacle polygons as obstacle information. If some obstacles are non-linear in shape, please give a discussion on it.

2. The reference list could be more expansive, covering the latest research results on maneuvering target tracking using the multiple-model estimator. For example:

[*]"Constrained dynamic systems: Generalized modeling and state estimation." IEEE Transactions on Aerospace and Electronic Systems, Oct. 2017.

[**]"Hybrid grid multiple-model estimation with application to maneuvering target tracking." IEEE Transactions on Aerospace and Electronic Systems, Feb. 2016.

3. I would like to see the time complexity comparison.

4. The calculation of the TPM and the MP depends on the fuzzy inference rule and please discuss how to reasonably design it. In order to further show the superiority of the proposed algorithm, it is better to present the performance comparison of the algorithms using different fuzzy inference rules.

Author Response

Response to Reviewer 2 Comments

This paper utilizes the distance and the field angle as the obstacle information and analyzes their relationship, and then maps them to a set of fuzzy rules, which will modify the TPM and MP for the state estimate. In my opinion, this topic is interesting. Nevertheless, addressing the following issues will improve the paper further.

Point 1: The proposed algorithm uses polygons to describe the obstacles and then extracts the distance and the field angle of these obstacle polygons as obstacle information. If some obstacles are non-linear in shape, please give a discussion on it.

Response 1:

In practical application, many obstacles have various shapes. However, in order not to add too much computational load, FOIA-MM algorithm simplified the shape of the obstacle into a polygonal shape. Concave polygon will produce abrupt changes in practical application, which will greatly affect the tracking performance and even reduce the tracking performance. If it is concave polygon, it must first be filled as convex polygon. Convex polygon used in this algorithm is enough to cover the shape of obstacles. For obstacles with non-linear shape, This paper use convex polygon, which can best cover its shape, to simplify. FOIA-MM algorithm will declare all polygon as convex polygons.

Corresponding modification:

At line 138-145 of page 5.

Point 2: The reference list could be more expansive, covering the latest research results on maneuvering target tracking using the multiple-model estimator. For example: [*]"Constrained dynamic systems: Generalized modeling and state estimation." IEEE Transactions on Aerospace and Electronic Systems, Oct. 2017.

[**]"Hybrid grid multiple-model estimation with application to maneuvering target tracking." IEEE Transactions on Aerospace and Electronic Systems, Feb. 2016.

Response 2:

This paper expands on current research and adds the two references.

Corresponding modification:

At line 35 of page 1.

At line 41of page 2.

At line 328-335 of page 14.

Point 3: I would like to see the time complexity comparison.

Response 3:

 I have modified and added a comparison of the time complexity of the two algorithms.

Corresponding modification:

At line 263-271 of page 11.

Point 4:The calculation of the TPM and the MP depends on the fuzzy inference rule and please discuss how to reasonably design it. In order to further show the superiority of the proposed algorithm, it is better to present the performance comparison of the algorithms using different fuzzy inference rules.

Response 4:

 Through the comparison of experiments, the fuzzy rules I used in this paper are the methods that can get better performance through adjustment at present. Fuzzy rules are difficult to adjust, and erroneous rules will be difficult to improve tracking performance. Adopting adaptive fuzzy rules is the research direction in the future.

Corresponding modification:

At line 298-302 of page 13.

Round  2

Reviewer 1 Report

I am satisfied with the paper . Thank you.